# Gender Identity, Disability, and Unmet Healthcare Needs among Disabled People Living in the Community in the United States

**DOI:** 10.3390/ijerph19052588

**Published:** 2022-02-23

**Authors:** Abigail Mulcahy, Carl G. Streed, Anna Marie Wallisch, Katie Batza, Noelle Kurth, Jean P. Hall, Darcy Jones McMaughan

**Affiliations:** 1Center to Improve Veteran Involvement in Care, Portland VA Healthcare System, Portland, OR 97239, USA; 2Section of General Internal Medicine, Boston University School of Medicine, Center for Transgender Medicine and Surgery, Boston Medical Center, Boston, MA 02118, USA; cjstreed@bu.edu; 3Juniper Gardens Children’s Project, University of Kansas, Lawrence, KS 66045, USA; annawallisch@ku.edu; 4Women, Gender, and Sexuality Studies, University of Kansas, Lawrence, KS 66045, USA; batza@ku.edu; 5Institute for Health and Disability Policy Studies, Life Span Institute, University of Kansas, Lawrence, KS 66045, USA; pixie@ku.edu; 6Research and Training Center on Independent Living and The Institute for Health and Disability Policy Studies, Lawrence, KS 66045, USA; jhall@ku.edu; 7School of Community Health Sciences, Counseling and Counseling Psychology, Oklahoma State University, Stillwater, OK 74078, USA; darcy.mcmaughan@okstate.edu

**Keywords:** disability, transgender, unmet health needs

## Abstract

Disabled adults and transgender people in the United States face multiple compounding and marginalizing forces that result in unmet healthcare needs. Yet, gender identity among disabled people has not been explored, especially beyond binary categories of gender. Using cross-sectional survey data, we explored the rates of disability types and the odds of unmet healthcare needs among transgender people with disabilities compared to cisgender people with disabilities. The rates of disability type were similar between transgender and cisgender participants with two significant differences. Fewer transgender participants identified physical or mobility disability as their main disability compared to cisgender participants (12.31%/8 vs. 27.68/581, *p* < 0.01), and more transgender participants selected developmental disability as their main disability compared to cisgender participants (13.85%/9 vs. 3.67%/77, *p* < 0.001). After adjusting for sociodemographic characteristics, the odds of disabled transgender participants reporting an unmet need were higher for every unmet need except for preventative services.

## 1. Introduction

People living with disabilities (following APA guidelines [1], we alternate between person-first and identity-first language throughout this paper) are a distinct and substantial historically and intentionally excluded population in the United States. An estimated 61 million community-dwelling adults in the US, more than 25% of the population, live with at least one disabling condition [2]. While “people living with disabilities” is often conceptualized as a single, homogeneous group, the reality is that disabled people are diverse. From the types of disability experienced, to the range of needs, to sociodemographic and economic characteristics, the disabled community is wide and varied. Disabled people may live with vision, hearing, psychiatric, developmental, cognitive, mobility, independent living or self-care disabilities—or any combination of multiple disabling conditions [3]. Two people with the same disability might be affected differently and have different needs based on income, social and community factors, genetics, access to and quality of healthcare, as well as race, ethnicity, and gender.

Disablism and ableism affects these factors and their impact on disability, often causing intersecting struggles [4] and unmet needs [5,6,7]. Using a nationally representative survey of people residing in the community in the United States (the Medical Expenditure Panel Survey), Reichard et al. found that among people with insurance, people with disabilities were more than two times as likely to report unmet needs compared to adults without disabilities [8]. The rates vary by disability subgroups. For example, youth and adolescents with disabilities experienced higher rates of unmet health needs compared to youth and adolescents without disabilities (8% compared to 2.8% in a study of the 2016 National Survey of Children’s Health) [9]. Among older people with disabilities, ableism and discrimination can be associated with unmet healthcare needs [10].

Transgender people also experience higher rates of unmet healthcare needs compared to cisgender people. James et al. found that 23% of the participants in the 2015 US Transgender Survey reported unmet healthcare needs [11]. In their sample of transgender veterans, Lehavot et al. report that almost half of their respondents (46%) indicated they delayed seeking medical care (outside the VA system) [12]. A sample of Black transgender women (*n* = 235) in two California cities (San Francisco and Oakland) also reported unmet healthcare service needs (16.6%) [13].

Additionally, people with disabilities face discriminatory employment environments [14], impoverishment [15], higher rates of arrests and incarceration [16,17], and stigma [18,19], as well as barriers to healthcare [7,20]. Some barriers include the prohibitive costs of care [21], a lack of health-adequate insurance coverage, a lack of transportation [22], personal and cultural barriers [22], and a lack of knowledge among medical professionals about disability [22,23]. These barriers, and others, lead adults with disabilities to postpone care or avoid the healthcare system altogether [8,22]. While disability status is not the same as health status, as a result of systemic issues in the healthcare system, adults living with disabilities are more likely to experience poor health outcomes compared to adults without disabilities. For example, people living with disabilities are more likely to experience cardiac disease, high blood pressure, high cholesterol, diabetes, stroke, arthritis, and asthma than people living without disabilities [24]. Disabled people are particularly excluded and marginalized during emergencies and disasters [25]. During the COVID-19 pandemic, healthcare policies and precautions to mitigate the spread of the virus led to disruptions in community-based care and clinical services [26]. That is, the prioritization of health education, access to medications, services, and life-saving equipment shifted away from disabled people [27,28], resulting in an increased risk for severe complications and death [29,30,31].

These inequities unduly affect people living with disabilities from groups that have been historically marginalized or made vulnerable. As a result, for example, disabled people of color may experience compounding adversities due to systemic inequities [5,32]. While disabled people have a higher probability of arrest compared to people without disabilities, Black disabled people have an even higher probability of arrest compared to whites with disabilities [17]. Black and Indigenous people living with disabilities have the highest poverty rates compared to all other people, with or without a disability [33]. Part and parcel of the intersectional discriminating forces oppressing disabled people in the United States is the lack of understanding of gender identity and sexual orientation of people living with disabilities. Gender identity, in particular, is woefully understudied in people living with disabilities. While inequities along the gender binary are relatively well documented, people without disabilities are often unwilling to acknowledge the gender identity rights of disabled people [34], leaving transgender, non-binary, or gender non-conforming people with disabilities silenced. Research that does focus on disability and gender identity does so by questioning transgender people about their disability status, rather than asking disabled people about their gender identity. In particular, even though research suggests there are higher rates of gender dysphoria among those with intellectual and developmental disabilities, especially autism spectrum disorder [35], the sexuality and gender identity of these individuals are often dismissed [36].

Understanding gender identity beyond the gender binary among disabled people is important because, as with other historically excluded and intersecting groups, disabled adults and transgender people in the United States face multiple compounding and marginalizing forces, including but unfortunately not limited to impoverishment, workplace discrimination, violence, interactions with the criminal justice system, housing discrimination, discrimination in healthcare and resulting health disparities, and marriage inequality [11,37,38,39,40,41]. Like adults living with disabilities, transgender adults are at a higher risk of multiple negative health outcomes. For example, the transgender population, compared to the cisgender population, is more likely to face HIV exposure, substance abuse, self-injury, and suicidality [42,43]. Likewise, transgender adults may also lack access to appropriate health services [44]. Barriers limiting access to care for transgender adults may include lack of insurance [45], fear around disclosing gender identity, structural barriers [46], and discrimination by healthcare providers [45,46]. When focusing on transgender populations, we find disability to be associated with stigma, discrimination, and psychosocial distress among transgender, non-binary, and gender non-conforming people. Disability is a risk factor for suicidal thoughts and behaviors among transgender and gender diverse adults [47]. In Australia, Leonard and Mann conducted an analysis of literature, policy, and survey data on the lived experiences of lesbian, gay, bisexual, transgender, and intersex (LGBTI) people with disability. They concluded that LGBTI people with a disability face discrimination, lack of access to services, social supports, and community connections (in both the disability and LGBTI communities) [34]. In the United States, transgender people with disabilities report higher rates of economic hardship, psychological distress, suicidal behaviors, and mistreatment by healthcare providers compared to transgender people without disabilities [11]. Having a disability is also associated with poly-victimization among transgender people [38]. Kattari et al. also found that disability was associated with an increased risk of victimization by healthcare providers, trans-related healthcare denials, and social service discrimination among transgender people [48,49,50]. Moreover, and perhaps, because of these barriers, transgender adults and those living with disabilities have unmet healthcare needs. For example, adults with disabilities report three times as many unmet healthcare needs as those without disabilities [51]. Among transgender adults in the United States, more than 32% reported unmet healthcare needs in the past year due to cost [52]. However, to our knowledge, no literature has focused on gender identity and unmet healthcare needs among people with disabilities.

Given both the lack of focus in the current literature on gender identity among people living with disabilities and the knowledge that transgender people living with a disability experience “avoidable, unnecessary, unfair and unjust” [53,54] inequities, we set out to understand gender identity, disability, and unmet healthcare needs among a sample of people with disabilities in the National Survey on Health and Disability (NSHD). The National Survey on Health and Disability is one of the few surveys that only collects information from disabled people and includes questions on gender identity. This is important because gender identity development is not encouraged among people with disabilities, thus disabled people—particularly those with intellectual or developmental disabilities—outside the gender binary might not be included in surveys of transgender people. Furthermore, the literature shows that access to care differs among groups within the population of disabled people [55,56]. However, to our knowledge, no studies exist that focus on unmet needs with gender identity as a disability subgroup. Thus, there is a need to conduct research to understand the experience of unmet need among people with disabilities who identify outside the gender binary. To address this gap in the literature we asked the following research questions:Which disabilities are most common among transgender people with disabilities?Do the rates of types of disabilities differ between transgender people with disabilities and cisgender people with disabilities?Are there differences in unmet healthcare needs based on gender identity among disabled people?

### 1.1. Transgender, Non-Binary, and Gender Non-Conforming Identities and Definitions

Transgender people are those whose gender identity does not align with sex assigned at birth. As a point of comparison, cisgender people are those whose gender identity does align with sex assigned at birth [57]. For this project, transgender is an umbrella term containing multiple distinct identities that include but are not limited to male-to-female, female-to-male, non-binary, genderqueer, and others [58]. Historically, transgender individuals are included in a larger population of lesbian women, gay men, bisexual people, queer people, intersex people, and others (LGTBQI*) [59].

### 1.2. Positionality Statement

The lived experiences of the authors lent to an interest in exploring the intersection between gender-identity and disability. More than one author is a disabled and non-binary academic. All are health services researchers with backgrounds in studying how gender, sexuality, and disability affect the cost, quality, and access to healthcare in the United States.

## 2. Materials and Methods

### 2.1. Study Design

This study was an observational and descriptive cross-sectional study of internet survey data from a convenience sample of disabled people living in the community in the United States.

### 2.2. Setting and Data

The data used for this study came from the 2019 wave of the National Survey on Health and Disability (NSHD), an internet-based survey fielded between October 2019–January 2020 and funded by the National Institute on Disability, Independent Living, and Rehabilitation Research (NIDILRR) [60].

### 2.3. Participants, Recruitment, and Privacy Protections

To be eligible for inclusion in the NSHD project, participants must have resided in the United States or a U.S. territory at the time of the study, be 18 years of age or older, and have a disability. Recruitment occurred through solicitations targeting national and state disability organizations, national listservs and newsletters, national conferences focusing on disability, and social media outlets using MTurk for recruitment (see Ipsen, Kurth, and Hall, 2021 [61], for details). Three screener questions were used to determine eligibility. The first two screeners confirmed residency and age. The third screener asked if the participant had a “physical or mental condition, impairment, or disability that affects daily activities or requires the use of special equipment or devices, such as a wheelchair, walker, TDD or communication device”. If participants answered yes to each of the three screeners they proceeded through the survey.

The 2175 respondents who participated in the survey were disabled people with a multitude of disability types between the ages of 18 and 62 years and residing in the United States. We obtained the de-identified data set with no personally identifiable information. The study was approved by the University of Kansas IRB. Unique identifiers for each participant were assigned by University of Kansas Institute for Health and Disability Policy Studies, and variables with cell sizes smaller than 20 were removed prior to receiving the data. Based on this, the project was deemed exempt from review by the Oklahoma State University Internal Review Board.

### 2.4. Measurement

The NSHD survey domains for this analysis included the following: (1) Demographics, (2) Disabilities, and (3) Unmet Healthcare Needs. In the following section we describe the items taken from each domain that were used in the analysis.

#### 2.4.1. Main Disability

Main disability was determined by asking participants “of the options listed below, which ONE category would you use to describe your main disability or health condition”. Seven categories were listed with the order randomized for each participant: intellectual or cognitive, mental illness or psychiatric, physical or mobility, chronic illness or disease, sensory, developmental, and neurological.

#### 2.4.2. Gender Identity

The measure for gender identity came from the Demographics domain in the NSHD. Participants were asked to identify as male, female, or other. Those who chose “other” were asked a follow up question that allowed them to identify as transgender, non-binary, two-spirit, gender non-conforming, genderqueer, agender, and intersex as well as the option not to disclose. Of those who identified as “other”, the majority identified as transgender and non-binary. Several identified as intersex or preferred not to disclose. Exact percentages of identities in the “other” category are hidden due to the small sample size. For the purpose of this study, we follow the example of others who argue that intersex falls under the transgender umbrella [11], and that it is more common for people with a gender identity that falls outside the range of the normative gender binary (i.e., transgender) to prefer not to disclose. We made a binary measure including cisgender (male/female) and transgender (other) for use in our analysis.

#### 2.4.3. Unmet Healthcare Needs

Unmet healthcare needs were measured using four items from the NSHD Community Participation Domain that asked participants who reported having health insurance about their unmet healthcare needs. Respondents were asked the following as a series of five separate questions based on type of healthcare: “in the past 12 months, have you been able to (see the doctors; get all the prescription medications; get preventative services; see the specialists; get the dental services) you need with your health insurance plan(s)?” The response options were “yes”, “no”, “I don’t know”, and “I did not need to (see a doctor; get prescription medication; see a specialist; get dental service)”. A participant was considered to have an unmet need if they answered “no” to any of the five questions. More details about these items and how they were developed are found in Hughes et al. (2021) [62].

We focused on unmet healthcare needs only among participants with health insurance for several reasons. While lack of insurance is one of the strongest predictors of unmet need, particularly among people with disabilities [8,63], since the implementation of the Affordable Care Act, the rate of being uninsured among disabled people has been relatively low [64]. This was true within our sample, as almost 90% of the sample carried health insurance and almost 90% of transgender people carried health insurance. Thus, we were interested in exploring unmet need and gender identity among disabled people with insurance. Insurance status was ascertained by asking participants if they had health insurance through an employer or union, purchased private health insurance directly, were on Medicare, were on Medicaid, were on Tricare, or received healthcare from the Indian Health Service. Those who did not indicate that they received health insurance from one of these were asked if they had no insurance.

#### 2.4.4. Sociodemographic Characteristics

Demographic data used in this analysis were based on items asking participants to report their age, race and ethnicity, level of education, and income. Age at the time of the survey was calculated based on participants’ month and year of birth and the survey timestamp. Participants were asked to indicate the highest level of education they completed and their current. Income level was designated based on percent of the Federal Poverty Level and calculated using the number of people in the respondent’s household, their reported income, and their state of residence. One item captured both race and ethnicity. This item instructed participants to select which one or more racial and ethnic categories best described their race and ethnicity. Participants could select multiple options. The race and ethnicity variables were calculated based on these responses to this item to produce a categorical (yes/no) variable for each race and ethnicity category. Categories included American Indian or Native American, African American or Black, Asian, Hispanic or Latino, Native Hawaiian or Pacific Islander, White or Caucasian, and Other. We included these variables to be in alignment with the body of literature associating these characteristics with unmet healthcare needs [51,62,65]. For example, a person’s ability to access dental healthcare is associated with race, ethnicity, education, and income [66].

### 2.5. Statistical Analyses

Data were modeled using tests of distribution and logistic regression. We examined univariate associations (STATA *crosstab*) between disability types and gender. Chi-square tests were used to determine statistical significance. We fit logistic regression models for categorical responses (STATA *logit*) to the data for each unmet healthcare need. The models included gender identity and were controlled for socio-economic and demographic characteristics (age, race, ethnicity, educational attainment, and income). Logistic regression analyses were conducted using Stata 15 [67]. Odds ratios (ORs) and 95% confidence intervals (Cis) are presented.

## 3. Results

In this section, we report, in both narrative and numeric form, the results and interpretation of our series of analyses exploring gender identity, disability, and unmet healthcare needs. The results are presented in multiple ways in keeping with the principles of Universal Design in Learning [68] to enhance accessibility by providing multiple ways of perceiving information. Table 1, Table 2 and Table 3 present the demographic characteristics of the 2175 disabled people who participated in the study, the rates of different types of main disability, and comparisons between cisgender and transgender, non-binary, and gender non-conforming participants. Each subsection below describes, after the narrative summary, the numeric details of participant characteristics, rates of main disability, and unmet healthcare needs by gender identity.

### 3.1. Narrative Summary of Results

As also reported elsewhere [69], overall, the adults living with disabilities who participated in the study were predominantly white, middle-aged, heterosexual, cisgender, women, and living with multiple disabling conditions. A little more than a quarter of the sample reported completing at least a four-year college degree, and almost a quarter reported completing a graduate degree. Thus, about half of the sample had a four-year college degree or higher. This rate is much higher than the educational levels reported in the literature. In the United States, people with disabilities are less likely to have completed a bachelor’s degree compared to people without disabilities [70]. Only 20% of disabled people aged 25 and over have a bachelor’s degree or higher, compared to 40% of those with no disability [71]. In our sample, 48% of the participants reported having at least a bachelor’s degree, which is 140% higher than the national average. Transgender, non-binary, and gender non-conforming people were a relatively small part of the sample, making up less than five percent of the total participants. However, this rate is higher than the rate of 0.55% reported in the general population [72]. Across all participants, the most reported main disability type was physical followed by mental illnesses or psychiatric disabilities, and the least commonly reported disability type was intellectual and developmental disabilities. Most participants carried insurance and reported multiple disabilities.

There were differences in the main disabilities between cisgender and transgender participants. Relatively few transgender participants selected physical or mobility disabilities as the single main disability type compared to cisgender participants. Transgender participants also chose developmental disabilities as the main type of disability at much higher rates than cisgender participants. Rates of mental illness or psychiatric disability, chronic illness or disease, neurological condition, sensory disability, and intellectual or cognitive disability were similar between transgender and cisgender participants. More transgender participants reported multiple disabilities, with almost 90% of the transgender, non-binary, and gender non-conforming participants living with more than one disabling condition, compared to half of the cisgender participants. Transgender and cisgender participants reported similar rates of health insurance at about 90%.

Even with insurance, we found that about 40% of the participants reported at least one unmet healthcare need. Looking at each type of unmet healthcare need, being unable to get all dental services needed was the most prevalent type of unmet need and being unable to get the preventative services as needed was the least prevalent type of unmet need. Looking at rates of unmet need by gender and disability type among people with health insurance, we found that more cisgender participants with a physical disability as their main disability reported at least one unmet need compared to transgender participants with a physical disability as their main disability. More transgender participants with a developmental disability as their main disability reported at least one unmet need compared to cisgender participants with a developmental disability as their main disability. Otherwise, rates of unmet need by disability type were similar between cisgender and transgender participants. However, after collapsing types of main disability and holding equal the sociodemographic characteristics we believed would affect experiences of unmet need, we still found that transgender disabled people were more likely to experience all types of unmet needs—except preventative services—compared to cisgender disabled people.

### 3.2. Study Participants Demographics

Participant demographics are found in Table 1. Most of the participants were white (78.9%/1716) and women (64.2%/1398). The median age was 40, with an interquartile range from 31 to 53 years old. Almost three percent of the sample (2.99%/65) identified as transgender, non-binary, or gender non-conforming.

### 3.3. Disability and Unmet Healthcare Needs across All Gender Identities

Rates of main disability and unmet healthcare needs across all gender identities are also found in Table 1. Across all participants, physical or mobility disability (27.08%/589) was the most selected main disability, followed closely by mental illness or psychiatric disability (26.30%/572) and chronic illness or disease (24.18%/526). Fewer participants selected neurological (10.8%/235), sensory (4.28%/93), developmental (3.95%/86), and intellectual or cognitive disabilities (2.90%/63).

Among respondents with health insurance (87.91%/1912), 44.24% reported at least one unmet healthcare need. Almost a third of the respondents experienced an inability to get all the dental services they needed (29.5%/561). The rates of unmet healthcare needs dropped for those unable to get prescription medications as needed (17.14%/326), unable to see a doctor as needed (12.72%/242), and unable to see a specialist as needed (12.36%/235). Fewer respondents reported being unable to get all the preventative services as needed (10.62%/202). A full description of unmet needs and disability among the study participants can be found in Hughes et al. (2021) [62].

### 3.4. Comparisons of Rates of Disability Types by Unmet Needs and Gender

Table 2 contains the unadjusted comparisons of rates of disability type and unmet need among transgender and cisgender participants. Looking at the self-selected main disability or health condition, transgender and cisgender participants reported similar rates of selecting intellectual or cognitive disability, mental illness or psychiatric, sensory disability, or neurological disability. About 30% of transgender (27.69%/18) and cisgender (26.39%/554) participants selected mental illness or psychiatric disability as their primary disability. A total of 10% of the participants, whether transgender (10.77%/7) or cisgender (10.86%/228), reported a neurological disability as the main disability. Sensory disability was the main disability for about 4% of the transgender participants (4.62%/3) and (4.29%/90) of the cisgender participants. Similarly, about 3% of the participants experienced intellectual or cognitive disabilities and their main disability or health conditions (3.08%/2) among transgender participants and (2.91%/61) among cisgender participants.

While many of the rates of disability type were similar between transgender and cisgender participants, there were two significant differences. Fewer transgender participants identified physical or mobility disability as their main disability compared to cisgender participants (12.31%/8 vs. 27.68/581, *p* < 0.01), and more transgender participants selected developmental disability as their main disability compared to cisgender participants (13.85%/9 vs. 3.67%/77, *p* < 0.001).

We also explored the rates of disability types by gender in the population with unmet needs. Rates of reporting disability type were similar between transgender and cisgender participants with unmet needs, with two exceptions: physical and developmental disabilities. The rate of cisgender people with unmet needs reporting physical disability as their main disability was greater than the rate of transgender participants with unmet needs who reported physical disabilities (27.40% vs. 10.53%, *p* < 0.05). Transgender people with unmet needs reported developmental disability as their main disability at a higher rate compared to cisgender people with unmet needs (15.79% vs. 3.93%, *p* < 0.001).

### 3.5. Rates and Odds of Unmet Healthcare Needs by Gender

Among participants with health insurance, the rates of reported unmet needs were higher among transgender participants compared to cisgender participants on every type of unmet need. At the highest rate, 47.27% of transgender respondents were unable to get all the dental care they needed, compared to 28.97% of cisgender respondents (*p* < 0.01). Almost a third (31.58%) of transgender people with disabilities reported being unable to see a specialist as needed, compared to just around a tenth (11.7%) of cisgender participants (*p* < 0.001). Around 40% of transgender participants (39.29%) endorsed being unable to see a doctor as needed compared to slightly more than a tenth (11.9%) of cisgender participants (*p* < 0.001). Just over 35% percent of transgender participants said they were unable to get prescription medication as needed (35.71%) compared to 16% (16.5%) of cisgender participants (*p* < 0.001).

After adjusting for sociodemographic characteristics, the odds of disabled transgender participants reporting an unmet need are higher for every unmet need except preventative services. By including each of these sociodemographic characteristics (race and ethnicity, age, income, and education) in our adjusted model we assess the relationship between gender and unmet need while holding other characteristics constant.

The transgender participants were four times more likely to report being unable to see a doctor as needed (OR = 4.12, 95% CI = 2.29–7.43, *p* < 0.001), and three times more likely to report being unable to get a prescription as needed (OR = 3.00, 95% CI = 1.67–5.40, *p* < 0.001) and all the dental services as needed, compared to cisgender participants (OR = 2.92, 95% CI = 1.63–5.22, *p* < 0.001). They were also 2.7 times more likely to experience an inability to see a specialist as needed (OR = 2.77, 95% CI = 1.50–5.11, *p* < 0.01).

## 4. Discussion

This study presents a look at disability and gender from a fresh angle and explores gender identity and disability differences among people known to have a disability from the National Survey on Health and Disability. Although the intersection of gender identity and disability has been interrogated for decades [39], we know little about gender identity prevalence among disabled adults. Much of this is due to the historical exclusion of gender identity from state and national surveys of health, hospital discharge data, and administrative health data [73]; likewise, the exclusion of transgender identity in nationally-representative surveys [74]. In our study, 3% of the disabled adults who participated in the NSHD were transgender, non-binary, and gender non-conforming. To our knowledge, there is no comparable literature focusing on gender identity among disabled people.

Unlike our study, where we compare rates of types of disability between disabled transgender and disabled cisgender participants, previous studies compared disability rates between transgender and cisgender people in a general sample of the population or from a sample of the population of transgender and non-binary people. For example, in their longitudinal study of veterans of the U.S. military, Brown and Jones found that transgender veterans had a higher rate of “catastrophic disability” compared cisgender veterans (8% vs. 2.9%), and were two times as likely to have a service-connected disability (OR 2.08) [75]. Among studies that use the 2015 United States Transgender Health Survey (*n* = 27,715 transgender, non-binary, and gender non-conforming participants) the rate of disability among transgender people is either 41.1% [47] or 28.47% [48], depending on how disability is operationalized. Downing and Przedworski compared transgender people to cisgender people on a variety of health-related outcomes using several years of the Behavioral Risk Factor Surveillance System, and found higher rates of disabilities among male-to-female people, female-to-male people, and gender non-conforming people compared to cisgender men and cisgender women [76]. These rates ranged from about 17% to 20% among transgender people compared to 5% to 15% among cisgender people. Rates of having multiple disabilities were also higher among transgender people compared to cisgender people, from 33% to 45% compared to 19% to 24%.

### 4.1. Prevalence and Importance of Disability Type among Disabled Transgender Participants

In focusing on gender identity among disabled people, we found that among disabled adults, transgender and cisgender people mostly experienced types of disability as the main disability at similar rates. For example, disabled transgender adults reported similar rates of mental illness and psychiatric disabilities compared to cisgender disabled adults. In our study, more than a quarter of the transgender respondents (27.6%) reported a mental illness or psychiatric disability as the main disability type, which is lower than rates of mental health issues reported in studies of transgender people (39%–59%), and higher than the rate of mental illness and serious mental illness in the U.S. population (20.6% and 5.2%, respectively) [11,77]. Our results do not necessarily line up with previous studies focusing on transgender and non-binary people, in which transgender study samples expressed much higher rates of mental illness and mental distress. Transgender veterans in Brown and Jones’s study were four times more likely to have a diagnosis of depression, twice as likely to have an eating disorder, and almost three times more likely to have post-traumatic stress disorder compared to cisgender veterans. Looking at the category of “serious mental illness” the transgender veterans in this study were over three times more likely to be diagnosed with one of these mental health conditions [11]. This may be due to participants in the National Survey on Health and Disability being asked to identify their main disability. Participants who did not select mental illness or psychiatric disability as their main disability type may still have a psychiatric diagnosis but that diagnosis is not their primary disability. Considering the possibility that transgender respondents in our study reported multiple disabilities, it is possible that the rate of having a psychiatric diagnosis is higher. Furthermore, the respondents reported higher than average educational levels and income levels, both of which are associated with fewer mental health issues and less exacerbation of existing mental health issues [78,79]. If mental health issues are less severe due to the stability and quality of life brought about by higher education and income, then respondents may be less likely to identify an existing mental health issue as their primary disability.

While disabled transgender and disabled cisgender participants reported similar rates of experiencing certain types of disability as the main disability (intellectual or cognitive disability, mental illness or psychiatric disability, sensory disability, and neurological disability), transgender, non-binary and gender non-conforming participants in our study were more likely to report a developmental disability as the main disability or health condition compared to cisgender participants. This aligns with other studies of neurodivergence and gender identity. In a large cross-section study of gender identity and neurodevelopmental traits among 641,860 people, Warrior et al. (2020) reported that the transgender and gender-diverse participants had higher rates of autism and other neurodevelopmental conditions compared to cisgender participants [80]. Using the National Survey on Health and Disability, Hall et al. (2020) found that participants who self-reported a diagnosis of autism also had high rates of self-reported LGTBQ+ identities. These autistic and LGTBQ+ participants experienced higher rates of mental illness, poor physical-health days, smoking, unmet healthcare needs, poor insurer provider networks, and being refused healthcare services [81,82].

### 4.2. Unmet Healthcare Needs among Disabled Transgender Participants

Overall, we found that transgender people with disabilities were almost three times more likely to report at least one unmet need compared to cisgender people with disabilities. This greater likelihood experienced by transgender people with disabilities is perhaps reflective of the compounding effects of both disability and gender identity on unmet healthcare needs.

People with disabilities experience higher rates of unmet healthcare needs compared to people without disabilities. Using a nationally-representative survey of people residing in the community in the United States (the Medical Expenditure Panel Survey), Reichard et al. found that among people with insurance, people with disabilities were more than two times as likely to report unmet needs compared to adults without disabilities [8]. The rates vary by disability subgroups. For example, youth and adolescents with disabilities experienced higher rates of unmet health needs compared to youth and adolescents without disabilities (8% compared to 2.8% in a study of the 2016 National Survey of Children’s Health) [9]. Among older people with disabilities, ableism and discrimination can be associated with unmet healthcare needs [10].

Compared to cisgender people, transgender people also experience higher rates of unmet healthcare needs. In this study, the prevalence of disability types differed for transgender and cisgender participants with unmet needs. Transgender participants with unmet needs were more than four times more likely than cisgender participants to report a developmental disability. In contrast, cisgender participants with an unmet need were more than two times as likely than transgender participants to report physical disability. Other studies have shown between 16% and 46% of transgender adults report some type of unmet need [11,12,13]. However, unmet healthcare service needs were operationalized differently in these studies compared to our current study. For example, in the Nemoto study on Black transgender women in California, healthcare services needs included sexually transmitted diseases screening, emergency room use, alternative healthcare, and a category called “other health services” [13]. From another study using the NSHD, sexual and gender minority persons with disabilities with and without health insurance reported numerous unmet healthcare needs [83]. Thus, it is important to highlight unmet needs by specific type of healthcare services, as the type of service measured can affect rates of unmet need.

#### 4.2.1. Unable to See a Doctor as Needed

The rates of unmet needs in primary care among people with disabilities across the U.S. population have dropped in recent years, from 47% in 2009 to 41% in 2014 [84]. The rates among our sample of disabled people with insurance are still lower. However, we found significant differences in rates of being unable to see a doctor as needed between transgender and cisgender people with disabilities, suggesting that transgender disabled people were almost 500% more likely to experience this unmet need. This suggests that although inroads have been made in addressing unmet healthcare needs for people with disabilities through targeted programs [84] and improvements in health insurance coverage [64], the gains are not experienced across all subgroups of disabled people. This may be due to primary care physicians lacking the education and experience needed to provide appropriate and accessible care to transgender people [57] and transgender people delaying or avoiding care due to negative experiences with care providers. For example, a third (33%) of the participants in the 2015 US Transgender Survey said they had at least one negative experience with a healthcare provider in the past year [11].

#### 4.2.2. Unable to Get All the Dental Services Needed

Oral health is a vital part of overall health [66]. Based on an analysis of the U.S. National Health and Nutrition Examination Surveys, an estimated 20% of the general U.S. population reported unmet dental care needs. The rates of unmet dental care needs vary by population subgroups, with 27.6% of Black people and 28% of Hispanic people reporting unmet dental care needs [85]. Among people with disabilities, there is a historically high burden of dental disease that remains unaddressed [86,87]. In our subgroup of transgender people with disabilities, almost half reported unmet dental needs, which was almost three times higher than the rate reported by cisgender disabled people. This is substantially higher than rates in the general population and rates reported in population subgroups. This is possibly due to an interaction between lack of disability accommodations in dental practices and a dental fear experienced by disabled people [88] and by transgender people that stems from negative and discriminatory behavior on the part of providers [89]. Dental care providers are also ill-prepared to provide appropriate care for people with disabilities, particularly people with intellectual and developmental disabilities [90].

#### 4.2.3. Unable to Get Prescription Medication as Needed

In our study, transgender disabled people were three times more likely to report an inability to get prescription medication as needed. This may be due to lack of access to hormone therapy, which is an important component of health and gender-affirming care [91,92]. Even with insurance, coverage of masculinizing hormone therapy can vary from 5% to 75%, and coverage of feminizing therapies can vary from 13% to 100% [93]. Lack of access and coverage can lead to sourcing hormone-related medications from unreliable sources, such as the internet [94,95].

However, considering the barriers to care, including costs, that transgender and disabled people experience, we should not assume that unmet needs among disabled transgender people center purely on gender-affirming therapies, especially since the literature is unclear around unmet prescription medication needs among disabled people. Recent research focuses on prescription medication misuse around pain medications and emphasizes findings that disabled people are more likely to misuse prescription medications [96]. However, the definition of misuse (“the use of prescription drugs without a prescription and also in any way a doctor did not direct respondents to use them” [96]) assumes an accessible and appropriate system of care for transgender people and people with disabilities. Both misuse (per this definition) and lack of access to prescription medications can overlap.

#### 4.2.4. Unable to See a Specialist as Needed

Access to specialty care is an ongoing issue in the United States. Community health clinics [97,98], safety-net hospitals [99], and rural areas [100,101] often lack specialty care services, and in settings with specialty care providers, the providers may be reluctant to accept public health insurance [98]. Transgender participants in our study were almost three times more likely to report being unable to see specialists as needed. While a lack of specialty care providers is problematic in the United States [101,102], lack of gender-affirming specialty care is a particular issue among transgender people [103]

Gender-affirming care is “healthcare that holistically attends to transgender people’s physical, mental, and social health needs and well-being while respectfully affirming their gender identity” [104]. In practice, specialty gender-affirming care consists of hormone therapy, “top”—chest reconstruction—and “bottom”—vaginoplasty, phalloplasty, metoidioplasty—surgery procedures, and puberty blockers [103]. Transgender individuals lack access to gender-affirming care and other appropriate health services [105]. Barriers limiting access to care for transgender adults may include insurance status [45], unwillingness to disclose gender identity, structural barriers [46], and discrimination by healthcare providers [45,46]. These barriers and others encourage some transgender adults to postpone healthcare [105].

Like gender-affirming care for transgender individuals, those with disabilities also struggle to access specialty care [106]. The largest issue in accessing specialty care for disabled individuals is a lack of adequate levels of health insurance [107]. Drainoni et al. (2006) found that issues navigating the health insurance system along with other factors obstruct or delay care for between 30% and 50% of disabled individuals [22]. The Patient Protection and Affordable Care Act addressed a number of issues for disabled individuals by putting an end to denial of coverage because of preexisting conditions, including a patient’s bill of rights that ended the lifetime cap on benefits, expanding the Medicaid program, and authorizing federally conducted or supported surveys and healthcare and public health programs to collect standard demographic characteristics that include disability status [107]. However, as not all states expanded Medicaid coverage, and only those that did expand coverage saw improved health insurance coverage for disabled individuals [108].

### 4.3. Limitations

The strengths of this study include the use of national data and a focus on disabled populations. However, there are several limitations to bear in mind while interpreting the results. The analysis is descriptive in nature, and while this provides a much-needed picture of gender identity among disabled people, it does not describe any outcomes of this relationship. The sample of transgender people in this study is small and heavily white. The way in which gender identity is measured may not adequately capture all non-cisgender participants. Differences between racial groups cannot be accurately assessed. Likewise, the disability types in the NSHD are self-reported so may be limited. The NSHD is an internet-based survey and carries with it the limitations of this design. These limitations include low response rate, self-selection bias, and problems with generalizing from an online population [109,110]. As a purposive convenience sample, the generalizability of results from the NSHD is limited. The population of transgender people in this study also most likely excludes those who are unhoused and incarcerated, biasing our results away from capturing the “true” rates of disability in the community, as people with disabilities are also more likely to experience homelessness and be justice involved. Furthermore, there is a risk that unmeasured variables associated with gender and unmet need may be driving the differences in unmet need between transgender and cisgender disabled people.

### 4.4. Implications

Our study highlights the importance of understanding gender identity and disability rates among disabled people and fills an important gap in the research. Existing research either focuses on gender identity and disability separately, frames disability as an outcome a transgender person may experience [111], or frames gender identity as a disability. In our study we explored gender identity among disabled people and found that disabled people who identify as transgender were more likely to identify developmental disabilities as the main disability compared to disabled cisgender people. This is important because historically, gender identities were not viewed as identities disabled people were capable of claiming [112,113], and transgender people were pathologized as morally corrupt and ethically disabled [114,115]. Additionally, due to this stigmatization, transgender people have, at times, resisted disability identities [116].

We found that transgender people with disabilities were more likely than cisgender people with disabilities to report unmet needs. Disabled people with an intersecting marginalized identity, such as a transgender, non-binary, or gender non-conforming identity may be at risk of compounding marginalization, discrimination, and exclusion. Fear of discrimination and stigma are often reasons transgender people delay healthcare and face unmet needs, and this delay can lead to worse health outcomes [117,118]. Disabled people with a non-normative gender identity need access to supports and services that support them as a disabled person and as a transgender, non-binary, or gender non-conforming person.

### 4.5. Future Research

Further data is needed to examine factors related to gender identity among disabled adults. Given the compounding effects of belonging to multiple historically excluded groups, particularly considering the systemic discrimination, stigmatization, and violence experienced by transgender people and disabled people, it is urgent and essential that we develop a better understanding of gender identity among disabled people. We need population data that are nationally representative and generalizable beyond the data sample. Rather than just creating new datasets, current population-level data that are nationally representative should start including items on disability and gender identity. Qualitative studies are needed to take a deep dive into the lived experiences of gender identity among disabled people, particularly around mental health-related disabilities and developmental disabilities. Additionally, and perhaps most importantly, community-engaged research that results in the co-production of knowledge around ways to eradicate compounded adversity faced by transgender disabled people is essential.

## 5. Conclusions

Transgender, non-binary, and gender non-conforming participants were more likely to report a developmental disability as their main disability compared to cisgender participants. Transgender people with disabilities were also more likely to have unmet healthcare needs compared to cisgender people with disabilities. These differences were particularly strong for the ability to see a doctor as needed and to get all dental services as needed. This greater likelihood of unmet needs experienced by transgender people with disabilities is perhaps reflective of the compounding effects of both disability and gender identity on unmet healthcare needs.

## Figures and Tables

**Table 1 ijerph-19-02588-t001:** Sample characteristics of the National Survey of Health and Disability (NSHD), 2019 (*n* = 2175).

Demographics	%(*n*) or Median (IQR)
Age	40 (31, 53)
Gender	
Cisgender	97.01% (2110)
Transgender	2.99% (65)
Race/ethnicity *	
American Indian or Native American	1.15% (25)
Hispanic/Latino	2.99% (65)
Black	4.47% (103)
Asian	2.21% (48)
Native Hawaiian or Pacific Islander	0.18% (4)
White	78.9% (1716)
Prefer not to answer	2.21% (48)
Level of Education	
No formal education	0.09% (2)
Less than high school	1.47% (31)
High school diploma	11.68% (254)
Some college	26.13% (568)
Two-year college	11.82% (257)
Four-year college	26.54% (577)
Graduate/Doctoral	21.43% (465)
Prefer not to answer	0.97% (21)
Income	
Under 138% FPL	37.00% (794)
138–249% FPL	22.55% (484)
250–399% FPL	18.59% (399)
400% FPL and above	21.85% (469)
Main Disability	
Mental illness or psychiatric disability	26.30% (572)
Chronic illness or disease	24.18% (526)
Physical disability	27.08% (589)
Neurological condition	10.80% (235)
Sensory disability	4.28% (93)
Intellectual or cognitive disability	2.90% (63)
Developmental	3.95% (86)
Has insurance	87.91% (1912)
Unmet need with health insurance ^+^	
Unable to see doctor as needed	12.72% (242)
Unable to get prescription medication as needed	17.14% (326)
Unable to see a specialist as needed	12.36% (235)
Unable to get all the dental services needed	29.50% (561)
Unable to get all the preventative services needed	10.62% (202)
Reported one unmet need	44.24% (903)
Reported multiple unmet needs	12.72% (242)

Note. Percentages based on non-missing responses. * Respondents could select more than one option. Percentages will not sum to 100%. ^+^ Percentages based on people with health insurance.

**Table 2 ijerph-19-02588-t002:** Comparisons between transgender and cisgender participants on rates of main disability type and unmet needs, NSHD 2019.

	Disability by Gender, % (*n*)			Prevalence of Disability Type by Gender in Those with at Least One Unmet Need ^+^, % (*n*)		
Disability	Transgender (*n* = 65)	Cisgender (*n* = 2110)	Chi2	Transgender (*n* = 38)	Cisgender(*n* = 865)	Chi2
Mental illness or psychiatric	27.69% (18)	26.26% (554)	0.796	23.68%(9)	24.39%(211)	0.921
Chronic illness or disease	27.69% (18)	24.08% (508)	0.502	26.32%(10)	26.71%(231)	0.958
Physical	12.31% (8)	27.54% (581) **	0.007	10.53%(4)	27.40%(237) *	0.021
Neurological condition	10.77% (7)	10.81% (228)	0.993	13.16%(5)	11.33%(98)	0.729
Sensory disability	4.62% (3)	4.27% (90)	0.891	7.89%(3)	3.47%(30)	0.155
Intellectual or cognitive	3.08% (2)	2.91% (61)	0.930	2.63%(1)	2.31%(20)	0.898
Developmental	13.85% (9)	3.65% (77) ***	0.000	15.79%(6)	3.93%(34) ***	0.001

Note. * *p* < 0.05; ** *p* < 0.01, *** *p* < 0.001 ^+^ Percentages based on people with health insurance.

**Table 3 ijerph-19-02588-t003:** Rates and odds of unmet need between transgender people with disabilities (*n* = 57) and cisgender people with disabilities (*n* = 1855) with health insurance, NSHD 2019.

	Rates by Gender, % (*n*)		Likelihood of Unmet Need of Transgender Compared to Cisgender People, OR (95% CI)	
Unmet Need	Transgender	Cisgender	Unadjusted	Adjusted ^+^
Unable to see doctor as needed	39.29% (22)	11.91% (220) ***	4.79 (2.75–8.33) ***	4.12 (2.29–7.43) ***
Unable to get prescription medication as needed	35.71% (20)	16.58% (306) ***	2.79 (1.60–4.90) ***	3.00 (1.67–5.40) ***
Unable to see a specialist as needed	31.58% (18)	11.7% (217) ***	3.46 (1.95–6.16) **	2.77 (1.50–5.11) **
Unable to get all the dental services needed	47.27% (26)	28.97% (535) **	2.20 (1.28–3.77) **	2.92 (1.63–5.22) ***
Unable to get all the preventative services needed	14.29% (8)	10.51% (194)	1.42 (0.66–3.04)	1.41 (0.65–3.08)
Reported at least one unmet need	66.67%(38)	43.6%(865) **	2.59 (1.48–4.52) **	2.70 (1.52–4.82) **

Note. ^+^ Adjusted for race, ethnicity, age, income, and education. ** *p* < 0.01; *** *p* < 0.001.

## Data Availability

Inquiries about the survey and dataset can be made by contacting the NSHD Administrator, Noelle Kurth, at pixie@ku.edu.

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
