# Peer review of "Gender Identity, Disability, and Unmet Healthcare Needs among Disabled People Living in the Community in the United States"

_ijerph, 2022, doi:10.3390/ijerph19052588_

Round 1
Reviewer 1 Report
Dear authors,
I found your study not only very interesting but very significant for the unmet healthcare needs in the disable people with variances in gender identity.The manuscript is well organised with quite extensive theory. Though, I would like to see some kind of data from Europe, Asia, etc. if possible. Sampling and methodology are appropriatel designed. This observatory, cross-sectional study suggests further research, as to my knowledge there is no other article to deal with these people. Well done!
Author Response
Reviewer 1
I found your study not only very interesting but very significant for the unmet healthcare needs in the disable people with variances in gender identity.The manuscript is well organized with quite extensive theory. Though, I would like to see some kind of data from Europe, Asia, etc. if possible. Sampling and methodology are appropriately designed. This observatory, cross-sectional study suggests further research, as to my knowledge there is no other article to deal with these people. Well done!
Authors’ reply: Thank you for your comments. At this time, we do not have access to comparable data from outside of the US. We look forward to exploring other data sources in future studies.
Reviewer 2 Report
Thank you for the opportunity to read this interesting paper about a topic I wasn't really aware of.
The paper is excellent and absolutely worth to be published.
Author Response
Reviewer 2
Thank you for the opportunity to read this interesting paper about a topic I wasn't really aware of. The paper is excellent and absolutely worthy to be published.
Authors’ reply: Thank you for taking the time to review our study. We appreciate your feedback.
Reviewer 3 Report
The authors of this study have put together an interesting piece of research that approaches the intersection of gender identity and disability in a novel way. This study contributes to our knowledge on intersectionality by looking at the unmet healthcare needs of people with disabilities who are also transgender, non-binary, and gender non-conforming, in doing a solid secondary, deqscriptive analysis of a national-level survey. On the whole it is also well-organized and well-written.
I have a some comments and criticisms that I offer in the spirit of trying to help improve the manuscript, as there are thoughts I had when reading, or things that might need some clarification.
In the category of gender identity, it seems like a lot has been collapsed together as “other.” I can accept that we're dealing with relatively small numbers to begin with, so that some collapsing needs to be done. I'm wondering if there's a better reference than #58 for putting intersex into transgender. Are there other sources that you've already used in the paper that can be pointed to on these recategorizations/groupings? For example, #59 (Hughes et al.) seems to be on the same survey. Or perhaps there's something with the 2015 Transgender Health Survey? I'm more than willing to accept the grouping, but just would like to see if there are any other studies that do something similar.
The authors have done well to provide good description of the socio-demographics of the survey respondents. While they mention some of the issues with who the responders would be in the limitations, I think more could be made of a lot of this. On page six, where they talk about the education levels of people that responded, they indicate that "this is slightly higher" then among people with disabilities more generally. That seems a vast understatement. The education levels for those with undergrad or graduate degrees is more than double the national average. Some more discussions reflecting this in that paragraph might be in order.
This might also be part of the explanation that they're trying to provide in section 4.1 where they say that the survey participants indicate far last mental health issues than what we know of the population. Education and income are always highly correlated, which means people have a better chance of a stable life, less stressors, and other things that might exacerbate mental health issues. As well, there are correlations between income and things like self-efficacy, so mental health might just be better overall because of the demographic.
It's an interesting finding that people who are Cisgendered with a physical disability tend to have one unmet health need, whilst people who are transgendered with the developmental disability also have an unmet need. This seems to disappear in the article right away, when the demographic characteristics are adjusted. Is that something that actually disappears quantitatively, or did it just go astray in the writing?
On page 9 the authors write that "After adjusting for socio-demographic characteristics" that transgendered respondents with disabilities come up higher in every category for unmet health needs. That's a key finding and it is useful to provide depth in the discussion on that as they have. As a reader though, I was left wondering what kind of adjustments? Can we get a line or two of discussion of what they did to norm that out?
I was struck that some of the literature that was introduced in the discussion, particularly at the start of section 4.2, and in section 4.2.3, should have been in the literature review. It seems to inform the critical questions that are being asked, but they're held back until here. I'm not sure if this is a strategic decision to hold back to emphasize the question, but it seemed strange to be introducing this literature new at the end of the study.
Author Response
Reviewer 3
The authors of this study have put together an interesting piece of research that approaches the intersection of gender identity and disability in a novel way. This study contributes to our knowledge on intersectionality by looking at the unmet healthcare needs of people with disabilities who are also transgender, non-binary, and gender non-conforming, in doing a solid secondary, deqscriptive analysis of a national-level survey. On the whole it is also well-organized and well-written.
I have some comments and criticisms that I offer in the spirit of trying to help improve the manuscript, as there are thoughts I had when reading, or things that might need some clarification.
- In the category of gender identity, it seems like a lot has been collapsed together as “other.” I can accept that we're dealing with relatively small numbers to begin with, so that some collapsing needs to be done. I'm wondering if there's a better reference than #58 for putting intersex into transgender. Are there other sources that you've already used in the paper that can be pointed to on these recategorizations/groupings? For example, #59 (Hughes et al.) seems to be on the same survey. Or perhaps there's something with the 2015 Transgender Health Survey? I'm more than willing to accept the grouping, but just would like to see if there are any other studies that do something similar.
Authors’ reply: We have replaced the reference to Dreger’s Galileo’s Middle Finger with a reference to the report from the 2015 United States Transgender Survey (USTS). The USTS used a similar definition of transgender identity. The final report includes more than 26 gender identities, including intersex, representing 3% respondents (see James, S. E., Herman, J. L., Rankin, S., Keisling, M., Mottet, L., & Anafi, M. (2016). The Report of the 2015 U.S. Transgender Survey. Washington, DC: National Center for Transgender Equality). We have followed suit and included intersex identity within our transgender population.
This can be found on page 5 in bold: “The measure for gender identity came from the Demographics domain in the NSHD. Participants were asked to identify as male, female, or other. Those that chose ‘other’ were asked a follow up question that allowed them to identify as transgender, nonbinary, two-spirit, gender non-conforming, genderqueer, agender, and intersex as well as the option not to disclose. Of those who identified as ‘other’, the majority identified as transgender and non-binary. Several identified as intersex or preferred not to disclose. Exact percentages of identities in the ‘other’ category are hidden due to small sample size. For the purpose of this study, we follow the example of others who argue that intersex falls under the transgender umbrella [58], and that it’s more common for people with a gender identity that falls outside the range of the normative gender binary (i.e. transgender) to prefer not to disclose. We made a binary measure including cisgender (male/female) and transgender (other) for use in our analysis”.
- The authors have done well to provide a good description of the socio-demographics of the survey respondents. While they mention some of the issues with who the responders would be in the limitations, I think more could be made of a lot of this. On page six, where they talk about the education levels of people that responded, they indicate that "this is slightly higher" than among people with disabilities more generally. That seems a vast understatement. The education levels for those with undergrad or graduate degrees is more than double the national average. Some more discussions reflecting this in that paragraph might be in order.
Authors’ reply: This is an excellent point. We addressed the much higher education levels of study participants.
On page 6, in bold: “This rate is much higher than educational levels reported in the literature. In the United States, people with disabilities are less likely to have completed a bachelor's degree compared to people without disabilities[70] . Only 20% of disabled people aged 25 and over have a bachelor’s degree or higher, compared to 40% of those with no disability[71]. In our sample, 48% of the participants reported having at least a bachelor’s degree, which is 140% higher than the national average.“
- This might also be part of the explanation that they're trying to provide in section 4.1 where they say that the survey participants indicate far last mental health issues than what we know of the population. Education and income are always highly correlated, which means people have a better chance of a stable life, less stressors, and other things that might exacerbate mental health issues. As well, there are correlations between income and things like self-efficacy, so mental health might just be better overall because of the demographic.
Authors’ reply: We concur. There is a potential that mental health may be better overall because of other demographic factors, like the higher educational levels reported by the study participants. This may also be due to being overshadowed by the main disability type. Participants in the NSHD were asked to identify their main disability type. It may be that participants with multiple disabilities did not choose to name their mental health disability as their main disability type, resulting in what appears to be a lower rate of psychiatric diagnoses.
Text that addresses this issue is found on page 11, in bold: “This may be due to participants in the National Survey on Health and Disability being asked to identify their main disability. Participants who did not select mental illness or psychiatric disability as their main disability type may still have a psychiatric diagnosis, but that diagnosis is not their primary disability. Considering the possibility that transgender respondents in our study reported multiple disabilities, it is possible that the rate of having a psychiatric diagnosis is higher. Furthermore, the respondents reported higher than average educational levels and income levels, both of which are associated with fewer mental health issues and less exacerbation of existing mental health issues [79,80]. If mental health issues are less severe due to the stability and quality of life brought about by higher education and income, then respondents may be less likely to identify an existing mental health issue as their primary disability.”
- It's an interesting finding that people who are Cisgendered with a physical disability tend to have one unmet health need, whilst people who are transgendered with the developmental disability also have an unmet need. This seems to disappear in the article right away, when the demographic characteristics are adjusted. Is that something that actually disappears quantitatively, or did it just go astray in the writing?
Authors’ reply: There are transgender and cisgender people with all seven disability types who have at least one unmet need. Among those with unmet need, transgender and cisgender participants with phsycial disabilities and developmental disabilities differed significantly We have updated the accompanying text for clarity.
Updated text on page 12, in bold: “In this study, prevalence of disability types differed for transgender and cisgender participants with unmet needs. Transgender participants with unmet need were more than four times more likely than cisgender participants to report developmental disability. In contrast, cisgender participants with unmet need were more than two times more likely than transgender participants to report physical disability”.
- On page 9 the authors write that "After adjusting for socio-demographic characteristics" that transgendered respondents with disabilities come up higher in every category for unmet health needs. That's a key finding and it is useful to provide depth in the discussion on that as they have. As a reader though, I was left wondering what kind of adjustments? Can we get a line or two of discussion of what they did to norm that out?
Authors’ reply: In Table 3 we provide both the unadjusted and adjusted odds of unmet health care needs of transgender disabled people with health insurance compared to cisgender disabled people with health insurance. The unadjusted odds is the odds of a transgender disabled person having an unmet need in that particular health care category compared to a cisgender disabled person without taking into consideration any other characteristics of the participants.
But, we know from the previous research on unmet health care needs that other characteristics do, in fact, affect unmet health care needs. For example, in a study of unmet health care needs in primary care in Greece published in IJERP, the authors found that costs, age, gender, and educational attainment were associated with unmet health care needs (see Pappa, E., Kontodimopoulos, N., Papadopoulos, A., Tountas, Y., & Niakas, D. (2013). Investigating unmet health needs in primary health care services in a representative sample of the Greek population. International journal of environmental research and public health, 10(5), 2017–2027. https://doi.org/10.3390/ijerph10052017). Income and education were also found to predict unmet health care need in Korea (see Hwang J. Understanding reasons for unmet health care needs in Korea: what are health policy implications? BMC Health Serv Res. 2018 Jul 16;18(1):557. doi: 10.1186/s12913-018-3369-2). Similarly, cost, gender and age were associated with unmet ambulatory care needs in Hungary (see Lucevic A, Péntek M, Kringos D, Klazinga N, Gulácsi L, Brito Fernandes Ó, Boncz I, Baji P. Unmet medical needs in ambulatory care in Hungary: forgone visits and medications from a representative population survey. Eur J Health Econ. 2019 Jun;20(Suppl 1):71-78. doi: 10.1007/s10198-019-01063-0)
In our study we included age, race and ethnicity, level of education, and income to be in alignment with the body of literature associating these characteristics with unmet health care needs among people with disabilities and transgender people (see McColl, M.A.; Jarzynowska, A.; Shortt, S.E.D. Unmet Health Care Needs of People with Disabilities: Population Level Evidence. Disabil. Soc. 2010, 25, 205–218, doi:10.1080/09687590903537406.; Hughes, P.; Wu, B.; Annis, E.; Brunelli, B.; Kurth, N.; Hall, J.; Thomas, K. Association of Inadequate Provider Networks with Unmet Need for Health Services and Self-Employment among People with Disabilities. J. Healthc. Poor Underserved 2021. Giblon, R.; Bauer, G.R. Health Care Availability, Quality, and Unmet Need: A Comparison of Transgender and Cisgender Residents of Ontario, Canada. BMC Health Serv. Res. 2017, 17, 283, doi:10.1186/s12913-017-2226-z). This aspect of our study only focuses on people with health insurance, so we do not control for health insurance status.
By including these sociodemographic characteristics, we ‘adjust’ our model by holding each of those characteristics constant, so that we could make a statement about gender and unmet healthcare needs outside the influence of those characteristics. It is the convention of our fields (health services research and public health) to simply refer to this as “after adjusting for sociodemographic characteristics” and to discuss the adjusted odds ratios and associated confidence intervals. However, we agree with the reviewer - regardless of convention, it would be beneficial to clarify this statement for the reader.
We added to page 9, in bold: “By including each of these sociodemographic characteristics (race and ethnicity, age, income, and education) in our adjusted model we assess the relationship between gender and unmet need while holding other characteristics constant.”
- I was struck that some of the literature that was introduced in the discussion, particularly at the start of section 4.2, and in section 4.2.3, should have been in the literature review. It seems to inform the critical questions that are being asked, but they're held back until here. I'm not sure if this is a strategic decision to hold back to emphasize the question, but it seemed strange to be introducing this literature new at the end of the study.
Authors’ reply: We agree that the introduction of some of the literature in 4.2 and 4.2.3 seemed strange. We have chosen to move most of the material into the literature review.
We moved the following to page 2, in bold:Using a nationally-representative survey of people residing in the community in the United States (the Medical Expenditure Panel Survey) Reichard and colleagues found that among people with insurance, people with disabilities were more than two times as likely to report unmet needs compared to adults without disabilities [8]. The rates vary by disability subgroups. For example, youth and adolescents with disabilities experienced higher rates of unmet health needs compared to youth and adolescents without disabilities (8% compared to 2.8% in a study of the 2016 National Survey of Children’s Health) [9]. Among older people with disabilities, ableism and discrimination can be associated with unmet health care needs [10].
Transgender people also experience higher rates of unmet healthcare needs compared to cisgender people. James and colleagues found that 23% of the participants in the 2015 US Transgender Survey reported unmet healthcare needs [11]. In their sample of transgender veterans, Lehavot and co-authors report that almost half of their respondents (46%) indicated they delayed seeking medical care (outside the VA system) [12]. A sample of black transgender women (n=235) in two California cities (San Francisco and Oakland) also reported unmet healthcare service needs (16.6%)[13].
Also, we amended the beginning of the following paragraph, also on page 2, in bold: “Additionally, people with disabilities face discriminatory employment environments [14], impoverishment [15], higher rates of arrests and incarceration [16,17], and stigma [18,19] as well as barriers to healthcare [7,20]. Some barriers include prohibitive costs of care [21], lack of health adequate insurance coverage, lack of transportation [22], personal and cultural barriers [22], and lack of knowledge among medical professionals about disability [22,23].
We also continued to include these references in the discussion section but have made a point of including Streed, et al’s (2021) article only in the discussion as it helps to put our project in context with others using the NSHD.
Reviewer 4 Report
The central purpose of this study was to explore unmet healthcare needs among transgender and cisgender adults (aged from 18 to 62 years old) with disabilities from the United States. The sample included 2,175 participants with a variety of disability types. The study is both descriptive and analytic in nature. The descriptive issue is the investigation of disability types among transgender people with disabilities. The analytic questions that this study examines include the following: (1) Do types and rates of disability differ between transgender and cisgender people with disabilities? (2) Do transgender and cisgender people with disabilities differ in their unmet healthcare needs? In addition to gender identity (transgender vs. cisgender), predictors included age, race and ethnicity, marital status, level of education, and income. Univariate associations between disability types and gender identity and unmet health needs and gender identity. Logistic regression was used to predict unmet healthcare needs. Results indicate that, compared with cisgender participants, transgender individuals were more likely than cisgender people to report a developmental disability and unmet health needs.
This study covers an important topic, as disabled transgender individuals are more likely to face marginalization and exclusion in access to health services than their cisgender counterparts. Due to the risk of discrimination, many transgender people have unmet needs that may result in negative health outcomes. The manuscript is well written, and the analyses make sense. However, the study sample is small (transgender respondents number only 65 individuals), and its geographical scope is limited. Moreover, the number of predictor variables is small, and the authors did not present any substantial argumentation concerning their inclusion or exclusion. It is highly possible that spurious variables may account for the detected relationships.
Author Response
Reviewer 4
The central purpose of this study was to explore unmet healthcare needs among transgender and cisgender adults (aged from 18 to 62 years old) with disabilities from the United States. The sample included 2,175 participants with a variety of disability types. The study is both descriptive and analytic in nature. The descriptive issue is the investigation of disability types among transgender people with disabilities. The analytic questions that this study examines include the following: (1) Do types and rates of disability differ between transgender and cisgender people with disabilities? (2) Do transgender and cisgender people with disabilities differ in their unmet healthcare needs? In addition to gender identity (transgender vs. cisgender), predictors included age, race and ethnicity, level of education, and income. Univariate associations between disability types and gender identity and unmet health needs and gender identity. Logistic regression was used to predict unmet healthcare needs. Results indicate that, compared with cisgender participants, transgender individuals were more likely than cisgender people to report a developmental disability and unmet health needs.
This study covers an important topic, as disabled transgender individuals are more likely to face marginalization and exclusion in access to health services than their cisgender counterparts. Due to the risk of discrimination, many transgender people have unmet needs that may result in negative health outcomes. The manuscript is well written, and the analyses make sense.
- However, the study sample is small (transgender respondents number only 65 individuals), and its geographical scope is limited.
Authors’ reply: The study sample is small. However, the NSHD was the best available data source for this project. There is a dearth of data sources including transgender identity (Patterson, J. G., Jabson, J. M., & Bowen, D. J. (2017). Measuring sexual and gender minority populations in health surveillance. LGBT health, 4(2), 82-105) as well as disability information, especially in publicly available data sources. For example, the Behavioral Risk Factor Surveillance System (BRFSS) has included sexual and gender minority-related (SO/GI) questions since 2014. However, only 19 states included this optional module in 2014. As of 2019, 32 states included the SOGI module (Statistical Brief Using Sexual Orientation, Gender Identity, Sex, and Sex-at-Birth Variables in Analysis, n.d.). Of the more than 400,000 BRFSS adults with disabilities included in the 2015 BRFSS, it is still unknown how many identified as Transgender (BRFSS Dataset | Disability Statistics: Rehabilitation Dataset Directory, n.d.).
- Moreover, the number of predictor variables is small, and the authors did not present any substantial argumentation concerning their inclusion or exclusion. It is highly possible that spurious variables may account for the detected relationships.
Authors’ reply: The reviewer brings up a valid point: there is a risk that unmeasured variables associated with gender and unmet need may be driving the differences in unmet need between transgender and cisgender disabled people. That is, the relationship between gender and unmet need appears to be correlated but is not and is instead merely due to both being related to some unmeasured variable.
From the literature, we currently understand that the main drivers of unmet health care need include health insurance status, financial considerations, and sociodemographic characteristics. For example, in a study of unmet health care needs in primary care in Greece published in IJERP, the authors found that costs, age, gender, and educational attainment were associated with unmet health care needs (see Pappa, E., Kontodimopoulos, N., Papadopoulos, A., Tountas, Y., & Niakas, D. (2013). Investigating unmet health needs in primary health care services in a representative sample of the Greek population. International journal of environmental research and public health, 10(5), 2017–2027. https://doi.org/10.3390/ijerph10052017). Income and education were also found to predict unmet health care need in Korea (see Hwang J. Understanding reasons for unmet health care needs in Korea: what are health policy implications? BMC Health Serv Res. 2018 Jul 16;18(1):557. doi: 10.1186/s12913-018-3369-2. PMID: 30012117; PMCID: PMC6048816.). Similarly, cost, gender and age were associated with unmet ambulatory care needs in Hungary (see Lucevic A, Péntek M, Kringos D, Klazinga N, Gulácsi L, Brito Fernandes Ó, Boncz I, Baji P. Unmet medical needs in ambulatory care in Hungary: forgone visits and medications from a representative population survey. Eur J Health Econ. 2019 Jun;20(Suppl 1):71-78. doi: 10.1007/s10198-019-01063-0. Epub 2019 May 17. PMID: 31102157; PMCID: PMC6544592.)
In our study we included age, race and ethnicity, level of education, and income to be in alignment with the body of literature associating these characteristics with unmet health care needs among people with disabilities and transgender people (see McColl, M.A.; Jarzynowska, A.; Shortt, S.E.D. Unmet Health Care Needs of People with Disabilities: Population Level Evidence. Disabil. Soc. 2010, 25, 205–218, doi:10.1080/09687590903537406.; Hughes, P.; Wu, B.; Annis, E.; Brunelli, B.; Kurth, N.; Hall, J.; Thomas, K. Association of Inadequate Provider Networks with Unmet Need for Health Services and Self-Employment among People with Disabilities. J. Healthc. Poor Underserved 2021. Giblon, R.; Bauer, G.R. Health Care Availability, Quality, and Unmet Need: A Comparison of Transgender and Cisgender Res-idents of Ontario, Canada. BMC Health Serv. Res. 2017, 17, 283, doi:10.1186/s12913-017-2226-z. ). Our study only focuses on people with health insurance, so we do not control for health insurance status.
This is bolded and included on page 5:
“Demographic data used in this analysis were based on items asking participants to report their age, race and ethnicity, level of education, and income.
We included these variables to be in alignment with the body of literature associating these characteristics with unmet health care needs [51,63,66]. For example, a person’s ability to access dental health care is associated with race, ethnicity, education, and income [67].”
We also acknowledge limitations associated with unmeasured variables in our limitations section, bolded and on page 13: Furthermore, there is a risk that unmeasured variables associated with gender and unmet need may be driving the differences in unmet need between transgender and cisgender disabled people.